# Fifty years of evidence on perinatal experience among refugee and asylum-seeking women in Organization for Economic Co-operation and Development (OECD) countries: A scoping review

**Marwa Ramadan**[1]*, **Hani Rukh-E-Qamar**[1], **Seungmi Yang**[2], **Zoua M. Vang**[1]

**1** Department of Sociology, McGill University, Montreal, Quebec, Canada, **2** Department of Epidemiology, Biostatistics and Occupational Health, McGill University, Montreal, Quebec, Canada

* marwa.ramadan@mcgill.ca

**Data Availability Statement:** All relevant data are within the paper and its Supporting Information file.

## Abstract

### Background

Members of the Organization for Economic Co-operation and Development (OECD) play a significant role in hosting and supporting refugees. Refugees and asylum seekers in OECD countries may face unique challenges in accessing perinatal healthcare. These challenges can impact their use of and experience with perinatal health services leading to poor maternal and infant outcomes. This scoping review describes the general trends in perinatal health research among refugees/asylum seekers in OECD countries over the past fifty years (1970 to 2021) as well as summarizes their perinatal experience.

### Methods

Databases including Embase and Medline were searched using relevant key words for "refugee/ asylum seeker", "perinatal ", and " OECD countries.". Articles were excluded if they only involved economic migrants or internally displaced persons, conducted in non-OECD countries, only assessed health behaviors and practices during pregnancy (e.g., smoking), or were published in a language other than English. The final list of articles included 82 unique studies.

### Results

In the 40 years between 1970 and 2009, very few studies (n = 9) examined perinatal health among refugees/ asylum seekers in OECD countries. However, an increasing trend was observed over the past decade. Early studies (1980 to 2009) focused more on traditional perinatal outcomes; however, from 2010 onwards, studies related to perinatal experience were more likely to emerge in the global health literature. Access to timely prenatal care remains a challenge with failure to address the root causes of the problem in several OECD countries including those with a long history of hosting refugees. The limited availability of

**Funding:** This research was supported by CIHR grant no. 159451 and a William Dawson Scholar Award to ZMV. The funders had no role in study design, data collection and analysis, decision to publish, or preparation of the manuscript.

**Competing interests:** The authors have declared that no competing interests exist.

interpretation services and the lack of a patient-centered approach to care have also interfered with the perceived quality of care. In addition, perceived isolation and the limited social support experienced by this vulnerable population have negatively impacted their perinatal experiences in several OECD countries.

## Conclusion

Refugee/asylum seekers in OECD countries face a number of challenges during the perinatal period. Policy changes and further research are needed to address access barriers and identify specific interventions that can improve their well-being during this critical period.

## Background

By the end of 2021, 89.3 million people around the world were forcibly displaced due to conflict, violence, persecution, or human rights violation. Of these, 27.1 million were refugees, and 5.8 million were asylum seekers [1]. According to the United Nations High Commissioner for Refugees (UNHCR), members of the Organization for Economic Co-operation and Development (OECD) hosted over 7.2 million refugees in 2021. Data also show that three of the top five refugee hosting countries in 2021 were OECD countries. Specifically, Turkey, Colombia, and Germany alone hosted 6.9 million refugees [2]. At the time of writing, the number of refugees hosted in OECD countries is expected to be even higher as more Ukrainian refugees continue to be displaced to neighboring European countries.

Women and girls constitute approximately 50% of any refugee, internally displaced people (IDP), or stateless population [3]. Displacement and disasters can increase the risk of adverse health outcomes for both mothers and newborns, including preterm babies, low birth weight, congenital anomalies, and early pregnancy loss [4–7]. Previous literature also showed that women with asylum seeker or refugee status generally have worse perinatal health outcomes than other immigrant groups and are likely to report suboptimal access, use, and experience of perinatal health services [8]. These poor pregnancy outcomes have been attributed to several factors, including women's socio-cultural and economic backgrounds as well as the poor quality of services they receive [9–12].

The three-delay model has been used to capture areas in the care-seeking pathway that results in increased deaths and poor reproductive health outcomes among women in general [13]. The three delays include: 1) Delay in deciding to seek care; 2) Delay in identifying and reaching a health facility that provides care; and 3) Delay in receiving appropriate care after reaching the facility. This model may be particularly relevant for refugee and asylum-seeking women due to the often traumatic predeparture experiences in their sending countries and their precarious statuses in host countries.

Previous reviews have primarily focused on documenting adverse pregnancy outcomes among immigrant women, including refugees and asylum seekers [8,14–16]. However, only a few studies have examined aspects related to their perinatal experience. For example, a comprehensive study by Heslehurst et al. [8] performed an umbrella review of perinatal health outcomes and care among refugees and asylum seekers, with some OECD countries represented in their sample. While the latter provided an insightful general overview of perinatal access and experiences among migrant women, the specific aspects of these experiences among refugees and asylum-seeking women in OECD countries are yet to be deeply explored.

Given the significant increase in humanitarian migration to OECD countries over the past decade, coupled with the organization's commitment towards immigrants and refugees as part of the sustainable development goal 3 [17,18], it is imperative from a policy perspective to understand the various challenges faced by this vulnerable group for better integration, planning, and allocation of resources. Therefore, this scoping review aims to expand upon the existing literature by describing the general trends in perinatal health research among refugees/asylum seekers in OECD countries from 1970 to 2021 and synthesize the specific findings related to their perinatal experiences over the specified period.

## Methods

In March 2022, we searched Embase and Medline databases using Ovid for peer-reviewed English articles published between 1970 and 2021. This interval was selected to provide a historical and comprehensive perspective on the evolution of perinatal experiences among refugees and asylum seekers in OECD countries. It facilitated the examination of trends in perinatal health research in the context of major global displacement events, as well as the progress in perinatal care and research methodologies over the past five decades. This extensive range also allowed us to explore a wide variety of research, uncovering patterns and presenting a thorough overview of perinatal experiences for refugees and asylum seekers in OECD countries. PICOS criteria included: *Population*: asylum seekers or refugees; *Intervention*: pregnancy; *Comparative group*: non-refugees, non-asylum seekers for quantitative studies; *Outcome*: perinatal experience and selected perinatal outcomes including maternal morbidities (gestational diabetes, gestational hypertension), maternal mortality, small for gestational age, large for gestational age, fetal and neonatal mortality, and preterm birth; *Study design*: quantitative, qualitative, mixed methods.

The search strategy involved a combination of three concepts: "refugee/asylum seekers", "perinatal outcomes", and "OECD countries". Keywords for each concept were developed based on literature search and database-specific subject headings. We included studies of all designs (except for literature reviews, opinion pieces, and conference abstracts) that involved refugee and/or asylum-seeking populations hosted in OECD countries. Articles were excluded if they only involved economic migrants or internally displaced persons, conducted in non-OECD countries, only assessed health behaviors and practices during pregnancy (e.g., smoking), or were published in a language other than English.

In this study, we chose to conduct a scoping review instead of a systematic review or meta-analysis because our primary objective was to map the existing evidence on perinatal health experiences among refugees and asylum seekers in OECD countries. Given the wide search range, the heterogeneous nature of the available evidence, and the complex aspects of perinatal experiences that involve moving beyond the quantitative examination of evidence, a scoping review was better suited to meet our research goals. We defined the perinatal period as the timeframe spanning pregnancy up to one year after childbirth. To ensure transparency and rigor, we adhered to the Preferred Reporting Items for Systematic Reviews and Meta-Analyses extension for Scoping Reviews (PRISMA-ScR) guidelines while reporting the results of this scoping review [19]. No registered protocol is currently available for this scoping review.

We utilized the Covidence platform to streamline the review process, including the removal of duplicates, title and abstract screening, and full-text screening [20]. Article screening was performed by two independent reviewers and discrepancies were resolved by discussion. A descriptive and narrative approach was used to analyze results and synthetize findings, following a series of steps:

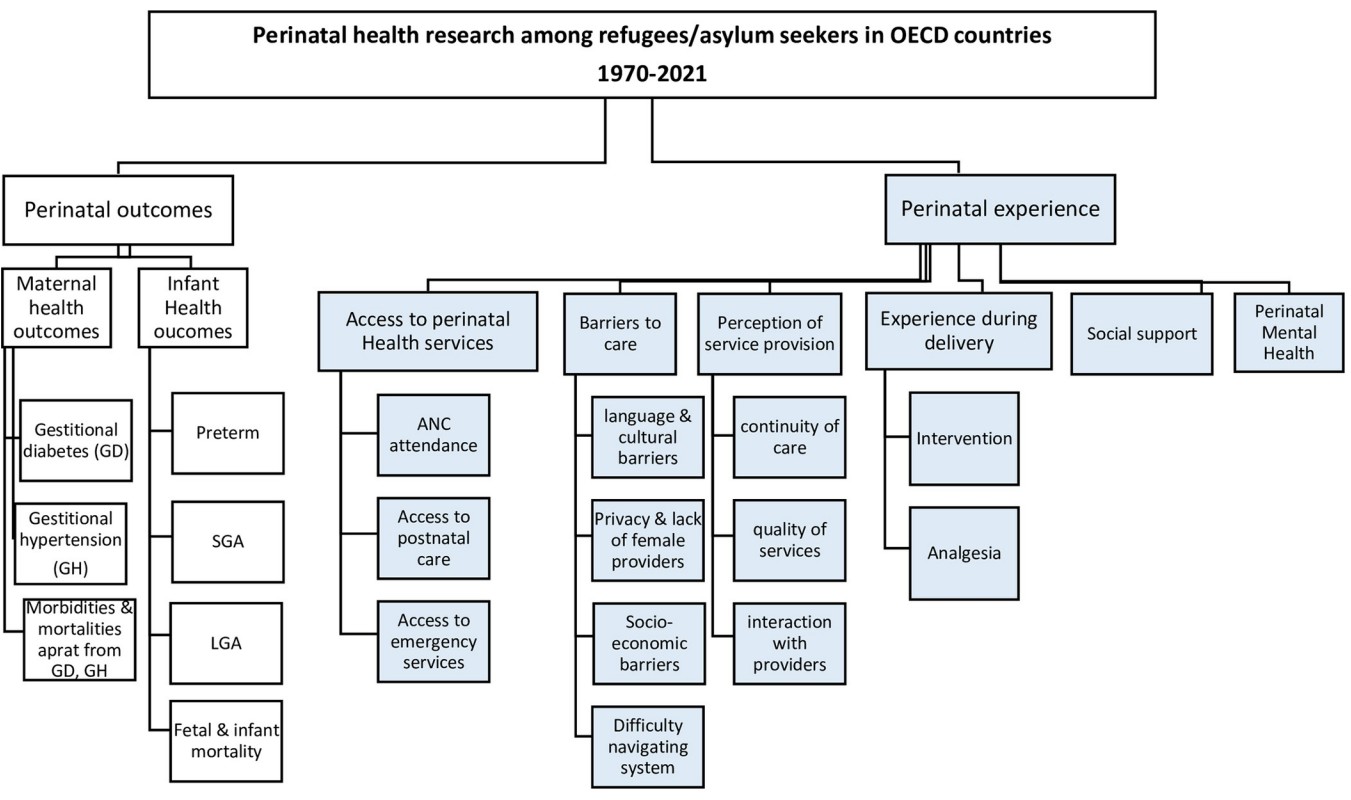

**Fig 1. Perinatal health research among refugees/asylum seekers in OECD countries 1970–2021.**

1. Data extraction: relevant data were extracted and systematically recorded using a standardized Excel spreadsheet that included information on study design, population, host country, intervention, comparator (for quantitative studies), outcomes, and key findings.

2. Data charting: our research team summarized and charted the extracted data to facilitate the synthesis of results.

3. Thematic analysis: two independent researchers coded the extracted data based on a predefined conceptual matrix, which was developed considering the themes and sub-themes identified in the 2020 Canadian list of perinatal health indicators [21] and additional perinatal experience themes of interest. The initial coding matrix was refined during the extraction process to incorporate new or additional sub-themes. The two researchers cross-checked the coded findings for consistency. Fig 1 shows the final coding matrix, including the breakdown of the themes and the sub-themes identified during the extraction process.

4. Descriptive analysis: we conducted a descriptive analysis to summarize the characteristics of the included studies and identify trends in perinatal health research in OECD countries, while considering global displacement crises.

5. Narrative analysis: we synthesized the results narratively, providing a qualitative summary and interpretation of the findings by describing common themes.

The final version of coding matrix consisted of two main categories: I) findings related to traditional perinatal health outcomes; and II) findings related to perinatal experience. Perinatal

health outcomes were further subdivided into two main themes: 1) maternal health outcomes; and 2) infant health outcomes. Prenatal experience was subdivided into six main themes: 1) Access to perinatal health services; 2) Barriers to care; 3) Perception of service provision; 4) Experience during delivery; 5) Social support; and 6) Perinatal mental health. Given that previous studies have focused on traditional perinatal outcomes, we only focused the narrative analysis in this review on themes related to perinatal experience and described their growing trends over the past 50 years compared to traditional perinatal outcomes.

## Results

The search process yielded 1146 articles, 979 of which were unique. Of these, 710 articles were excluded during title and abstract screening, and 269 articles were retained for full-text screening. Among the fully screened articles, 82 were included for final analysis, while 187 were excluded due to different outcomes from the defined (e.g., family planning, sexually transmitted infections, vertical transmission of Hepatitis B, congenital infections, abortions; n = 113), inappropriate patient population (n = 35), wrong study type (e.g., systematic reviews, opinion pieces or commentaries; n = 35), wrong setting/ non-OECD host country (n = 02), and not published in English (n = 02). Fig 2 illustrates the PRISMA flow diagram for this scoping review.

The final list of articles included perinatal findings among refugees/asylum-seeking women in 19 OECD countries between 1977 and 2020. Almost two thirds of included articles were quantitative (66%, n = 54), less than one third were qualitative (28%, n = 23), and 6% were of a mixed methods study design (n = 05). Most studies reported perinatal findings among refugees (82%, n = 67), one-fourth of the studies included findings for the asylum-seeking population (24%, n = 20), and four studies grouped findings from refugees and/or asylum seekers with other immigrant groups (e.g., economic migrants). A comparison group was specified in 35 out of the 54 quantitative studies. Specifically, 29 studies compared refugees/asylum seekers to a host or native-born population, while six studies compared them to other immigrant groups. Most studies were conducted in Australia (18%, n = 15), Turkey (17%, n = 15), Canada (16%, n = 13), Unites States (12%, n = 10), and Germany (10%, n = 8). Refugees /asylum seekers originated from various geographic regions, but the most common countries of origin included Syria (n = 22), Afghanistan (n = 11), Somalia (n = 11), Myanmar/Burma (n = 10), Sudan (n = 10), Eritrea (n = 10), Iraq (n = 9), and Ethiopia (n = 8).

In the 40 years between 1970 and 2009, very few studies (n = 9) examined perinatal health among refugees/ asylum seekers in OECD countries. However, an increasing trend has been observed over the past decade, coinciding with several global refugee crises. For example, as of July 2022, Syria was the largest global refugee crisis with more than 6.9 million Syrian refugees having fled their country since 2011. Among the Syrian refugees, 3.6 million (52%) reside in neighboring OECD countries such as Turkey 620,000 were resettled in non-neighboring OECD countries like Germany [22]. This pattern of displacement was clearly reflected in the results of this review with 22 studies (out of the 36 studies specifying the country of origin) having included data on Syrian refugees, and both Turkey and Germany are among the top five OECD countries with studies on refugee/asylum-seeking women. In addition, more than 60% of studies included in this review were published in the five years between 2017 and 2021 (n = 50) following the settlement of Syrian refugees in OECD countries.

In addition, we observed that early studies from 1980 to 2009 tended to focus more on traditional perinatal outcomes; however, from 2010 onwards, studies related to perinatal experience and mental health were more likely to emerge in the global health literature. This parallels the rising trend of studies on refugee/asylum-seeking women as well as the recent increase in global emphasis on perinatal mental health [23].

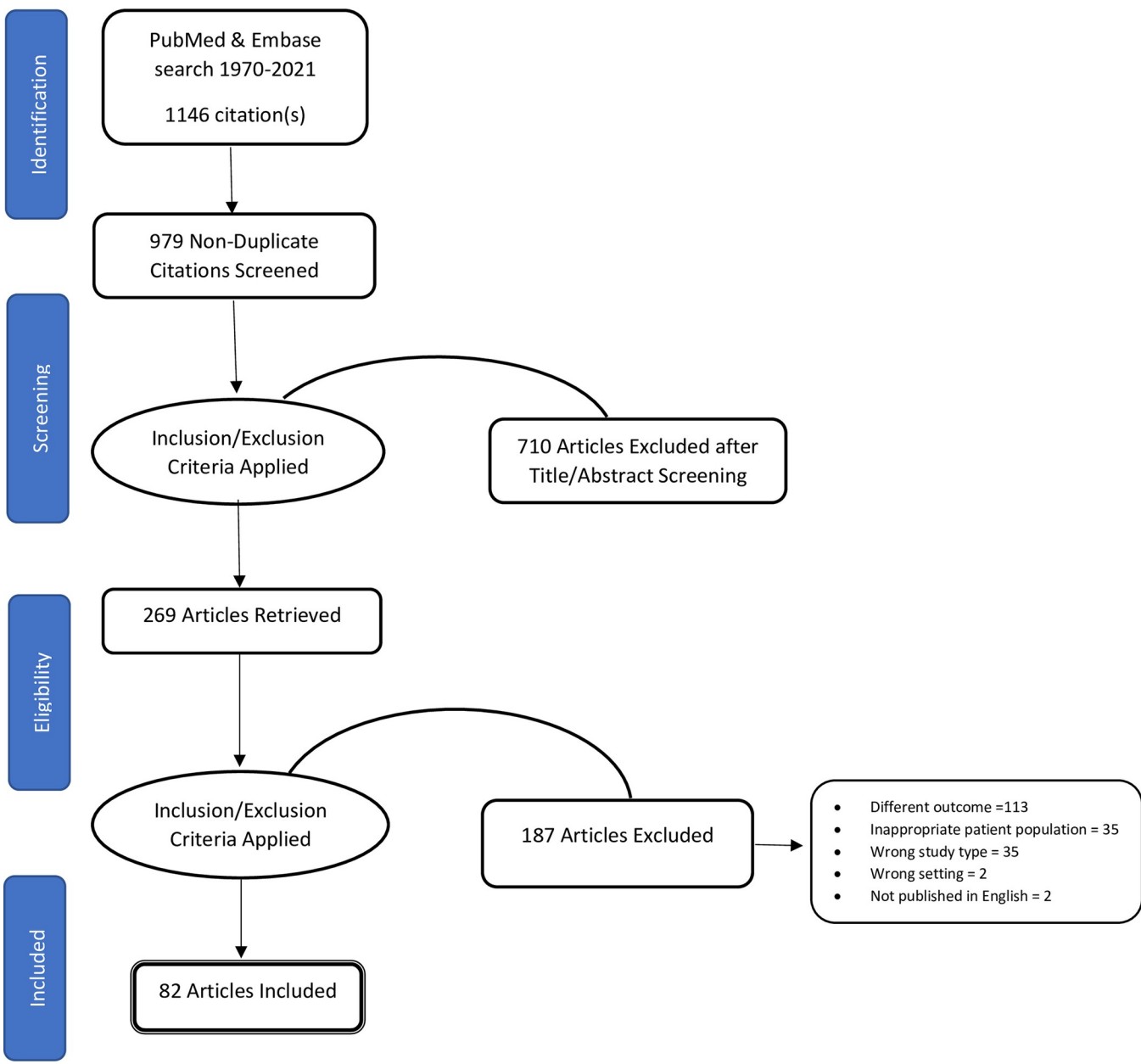

**Fig 2. PRISMA flow diagram for the scoping review process.**

## Summary of perinatal experiences

The majority of the studies in this scoping review included one or more sub-themes related to perinatal experience among refugee/asylum-seeking women in OECD countries (74%, n = 61). Although perinatal experience was reported in the global health literature as early as 1981, only in the past decade has more efforts been directed towards this area of research. Table 1 provides an overview of the quantitative studies, while Table 2 summarizes the qualitative and mixed methods studies that cover one or more subthemes related to perinatal experience among refugees/asylum-seeking women in OECD countries.

**Table 1.** List of quantitative studies on perinatal experiences of refugee/asylum-seeking women in OECD countries.

| Title | Study | Refugees | Asylum seekers | Host Country | Sample Size | Summary of Findings |
|---|---|---|---|---|---|---|
| Reproductive Health Outcomes in African Refugee Women: A Comparative Study. | Agbemenu 2019 | x | | United States | 789 African refugee | The majority of women started prenatal care in the first trimester, but refugees tended to initiate care later, in the second trimester, compared to US-born women. Refugees also have lower prenatal care adequacy scores compared to US-born women. Refugee women experience higher rates of vaginal deliveries and lower rates of C-sections, with fewer medical indications, compared to US-born women. |
| Perinatal Data of Refugee Women from the Gynaecology Department of Charite University Hospital Berlin Compared with German Federal Analysis | Ammoura 2021 | x | | Germany | 907 refugee women | There was no significant difference in C-section rates between the two groups, but the most common reason for a C-section, a previous C-section, was twice as prevalent among refugees. Pregnant women frequently reported experiencing psychological distress during pregnancy. |
| Satisfaction with maternity care among recent migrants: an interview questionnaire-based study. | Bains 2021 | x | | Norway | 401 International migrant women | Some women struggled to comprehend the information provided by healthcare professionals. Overall satisfaction with care was high, but women with a Norwegian partner, higher education, and strong Norwegian language proficiency were more likely to be dissatisfied. Refugee women sometimes felt they experienced differential treatment due to factors such as religion, language, and skin color, especially when compared to women who migrated for family reunification. |
| Pregnancy related health care needs in refugees-a current three center experience in Europe | Dopfer 2018 | x | | Germany | 2911 refugees | Pregnant refugees expressed a higher demand for care compared to pregnant migrants. Language barriers hindered communication and the ability to convey complaints among pregnant women. Some pregnant women preferred female doctors and refused care from male doctors, even when they were the only available option. |
| Clinical characteristics and pregnancy outcomes of Syrian refugees: a case-control study in a tertiary care hospital in Istanbul, Turkey. | Erenel 2017 | x | | Turkey | 300 Syrian refugees | Syrian refugees had significantly lower antenatal care (ANC) attendance compared to Turkish citizens, with 44% of refugees not receiving ANC care, compared to 7.7% among Turkish residents. Induction and cesarean section rates were considerably lower among Syrian refugee women (nulliparous) compared to Turkish residents. |
| Maternal and infant outcomes among Caucasians and Hmong refugees in Minneapolis, Minnesota. | Erickson 1987 | x | | United States | 350 Hmong refugees | Hmong mothers initiated prenatal care later than their Caucasian counterparts, with no information available on the number of antenatal care visits. |
| International migration to Canada: the post-birth health of mothers and infants by immigration class. | Gagnon 2013 | x | x | Canada | 1127 refugee, asylum seeker, immigrants) | Migrants were less likely to have their postpartum concerns professionally addressed, with issues ranging from psychological, such as self-harm and postpartum depression, to physical, like vitamin D supplementation for women. |

*(Continued)*

**Table 1.** (Continued)

| Title | Study | Refugees | Asylum seekers | Host Country | Sample Size | Summary of Findings |
|---|---|---|---|---|---|---|
| Maternal health and pregnancy outcomes among women of refugee background from African countries: a retrospective, observational study in Australia. | Gibson-Helm 2014 | x | | Australia | 1361North Africa =, Middle and East Africa = 706 and West Africa (n = | No differences were observed in the timing of the first antenatal care (ANC) visit and the adequacy of ANC between participants in (Humanitarian Source countries) HSC & non-HSC groups from North Africa, Middle East Africa, and West Africa. At least half of the women had their first visit after the first trimester. North African non-HSC women were more likely to undergo induced labor, episiotomies, and C-sections. For Middle & East African women, there was no difference between both groups, while West African non-HSC women had higher C-section rates. North African and Middle & East African non-HSC women were more likely to receive analgesia during labor, whereas there was no difference in analgesia rates between the two groups for West African women. |
| Maternal health and pregnancy outcomes among women of refugee background from asian countries | Gibson-Helm 2014 | x | | Australia | 2324humanitarian source countries (HSCs and 13976 non- HSCs | Humanitarian Source Countries (HSC) demonstrated higher rates of late booking and lower antenatal care attendance compared to non-HSC South Asian individuals, with no significant differences observed for Southeast or West Asian populations. Additionally, non-HSC South Asian individuals had significantly higher rates of labor induction, assisted vaginal delivery, and cesarean deliveries. |
| Maternal health and pregnancy outcomes comparing migrant women born in humanitarian and nonhumanitarian source countries: a retrospective, observational study. | Gibson-Helm 2015 | x | | Australia | 2,713 humanitarian source countries (HSCs) and 10,606 non-HSCs | A higher percentage of HSC (humanitarian source country) women had late booking and poor ANC attendance compared to non-HSC women. Additionally, HSC women experienced lower rates of induced labor, analgesia, C-section, and assisted birth. |
| Challenges in migrant women's maternity care in a high-income country: A population-based cohort study of maternal and perinatal outcomes. | GuÃ˚mundsdÃ³ttirEÃ 2021 | x | x | Iceland | | Migrant women in the lowest HDI group had higher odds of emergency C-section compared to Icelandic women. They also experienced higher rates of instrumental births and episiotomies, but lower rates of induction. |
| Health of Infants Born to Venezuelan Refugees in Columbia. | Guarnizo-HerreÃ±o 2021 | x | | Columbia | 1,947,810 Colombia and 5147 Venezuela refugees | The number of prenatal visits for Venezuelan refugees was half that of Colombian women. Most Venezuelan women did not have health insurance and faced barriers to accessing care |
| Pregnant women at risk: an evaluation of the health status of refugee women in Buffalo, New York | Kahler 1996 | x | | United States | 59 refugees | A high percentage of refugee women had their first ANC visit during the second or third trimester of pregnancy, with many not seeing a physician during their first pregnancy. At least one woman required a C-section. Many participants witnessed violence, some experienced rape while fleeing, and depression was observed in most of the study sample. |

*(Continued)*

**Table 1.** (Continued)

| Title | Study | Refugees | Asylum seekers | Host Country | Sample Size | Summary of Findings |
|---|---|---|---|---|---|---|
| Obstetric Outcomes of Syrian Refugees and Turkish Citizens. | Kanmaz 2019 | x | | Turkey | 4802 Syrian refugees, 6752low-income Turkish citizens, and 5446 high-income Turkish citizens | Refugees had significantly lower ANC attendance compared to Turkish citizens, with a higher proportion of refugees not receiving any ANC care. C-section and primary c-section rates were significantly higher among Turkish citizens compared to refugees. |
| Maternal mental healthcare needs of refugee women in a State Registration and Reception Centre in Germany: A descriptive study | Kaufmann 2022 | x | | Germany | 120 refugees | refugee women expressed fears related to pregnancy and delivery. A majority of patients were diagnosed with post-traumatic stress, adjustment, or depressive disorders, and 70% reported mental health issues during pregnancy. |
| Comparison of delivery characteristics and early obstetric outcomes between Turkish women and Syrian refugee pregnancies. | Kiyak 2020 | x | | Turkey | 940 Turkish women and 616 Syrian women | There was no difference in the number of women who did not receive antenatal care (ANC) between Turkish citizens and Syrian refugees. However, a significantly higher number of Syrian refugees chose to receive care at a tertiary healthcare center compared to Turkish citizens. Cesarean section rates were not different between the two groups. |
| Perinatal health of refugee and asylum-seeking women in Sweden 2014–17: a register-based cohort study. | Liu 2019 | x | x | Sweden | 31897 migrant women | Migrant women had higher rates of insufficient antenatal care (ANC) visits, with asylum seekers and undocumented migrants having more inadequate ANC, including no ultrasound screening, compared to refugees. Migrant women also had higher rates of missing postpartum visits compared to Swedish women, with asylum seekers and undocumented migrants more likely to miss these visits than refugee women. Additionally, migrant women had higher rates of induction and cesarean sections compared to Swedish-born women. Asylum seekers and undocumented migrants were more likely to have planned cesarean sections compared to refugees. |
| Born into direct provision: Outcomes of infants born to asylum seekers | Murphy 2020 | | x | Ireland | 78 asylum seekers | The gestational age at booking was quite late, around 30+4 weeks. There was no significant difference in the rates of cesarean sections or interventions during delivery between asylum seekers and hospital rates. Language barriers led to several miscommunications, with interpreters being used in only 20% of cases. Even when a need for an interpreter was identified, they were not consistently used at each appointment, leading to inconsistent reporting of maternal or obstetric history and challenges in delivering bad news. |
| Postpartum Pain in the Community Among Migrant and Non-migrant Women in Canada. | Mahon 2017 | x | x | Canada | 1022 International migrant women and 514 non-migrant women | Migrant women were more likely to begin antenatal care after three months due to difficulties in accessing services. There was no difference in epidural rates between migrants and non-migrants, and migrant women were less likely to have a regular healthcare provider. |

*(Continued)*

**Table 1.** (Continued)

| Title | Study | Refugees | Asylum seekers | Host Country | Sample Size | Summary of Findings |
|---|---|---|---|---|---|---|
| Antenatal Care Utilization and Obstetric and Newborn Outcomes Among Pregnant Refugees Attending a Specialized Refugee Clinic. | Malebranche 2020 | x | x | Canada | Total cohort = 179 [GAR = 78, PSR = 69, Asylum Seeker = 32] | A larger proportion of asylum-seeking women received inadequate antenatal care. No difference was found in obstetric and newborn outcomes with other refugee groups |
| Association Between Chronic Medical Conditions and Acute Perinatal Psychiatric Health-Care Encounters Among Migrants: A Population-Based Cohort Study | McKnight 2020 | x | | Canada | 29,189 refugees, (187,430 nonrefugee immigrants, and 641,385 long-term residents | Refugees with chronic medical conditions had a higher risk of experiencing acute perinatal psychiatric health issues compared to long-term residents. However, no significant difference was observed between non-refugee immigrants and long-term residents. |
| Obstetric Outcomes among Syrian Refugees: A Comparative Study at a Tertiary Care Maternity Hospital in Turkey | Ozel 2018 | x | | Turkey | 576 Syrian refugees and 576 ethnic Turkish women | The rate of prenatal care follow-up was significantly lower among refugees compared to other groups. While there was no difference in the type of delivery, the primary cesarean section rate was notably lower among the refugee group. |
| The influence of migration on women's use of different aspects of maternity care in the German health care system: Secondary analysis of a comparative prospective study with the Migrant Friendly Maternity Care Questionnaire (MFMCQ). | Seidel 2020 | x | | Germany | 184 nonimmigrant women, 214 immigrant women, and 62 direct descendants of immigrants. | On average, refugees started prenatal care in the 12th week of gestation, with only 63% beginning in the first trimester compared to 88% of German residents. A lower percentage of refugee women (53%) had more than nine prenatal care visits compared to German women (68%). Immigrant women generally had fewer than nine prenatal care visits and were less informed about postpartum care availability compared to German citizens. Language barriers and low income were significant factors affecting access to prenatal care and knowledge about postpartum care. Despite these challenges, Germany offered adequate prenatal care for both refugees and citizens. |
| Perinatal psychosocial assessment of women of refugee background. | Snow 2021 | x | | Australia | 100 women of refugee background and 100 Australian-born women w | The uptake of postnatal mental health (MH) referrals for women of refugee background was nearly double that of antenatal care (100% vs. 50%). This discrepancy may be due to barriers during antenatal MH care, as postnatal referrals were offered while in the hospital after giving birth. Although refugee women reported more psychological adversities during pregnancy compared to Australian women, they reported lower rates of mental health illness. |
| Childbirths and the prevalence of potential risk factors for adverse perinatal outcomes among asylum seekers in the Netherlands: A five-year cross-sectional study | Tankink 2021 | | x | Netherlands | | late perinatal care seeking, and the majority gave birth 2 months after arrival |
| The effect of cultural and linguistic diversity on pregnancy outcome | Thomas 2010 | x | | Australia | 4751 women of refugee background and those in need of interpretation service | use of interpreter service among women was associated with less likelihood of adverse pregnancy outcomes |

(*Continued*)

**Table 1.** (Continued)

| Title | Study | Refugees | Asylum seekers | Host Country | Sample Size | Summary of Findings |
|---|---|---|---|---|---|---|
| Severe acute maternal morbidity in asylum seekers: A two-year nationwide cohort study in the Netherlands | VanHanegem 2011 | | x | Netherlands | 1310 Asylum seekers and 369711 Dutch | Asylum seekers were more likely to have had a previous C-section compared to other groups. A majority of women (91.5%) faced communication challenges, with over half experiencing major language barriers or no communication at all. Family and friends were primarily utilized for interpretation services. |
| Pregnancy outcomes in asylum seekers in the North of the Netherlands: a retrospective documentary analysis. | VerschuurenAEH 2020 | | x | Netherlands | 344 Asylum-seeking women and 2323 Dutch women | On average, Asylum-seeking women began receiving antenatal care (ANC) after 13 weeks of gestational age (GA). Compared to Dutch women, refugee women experienced fewer inductions and received more analgesia during childbirth. |
| Are Syrian refugees at high risk for adverse pregnancy outcomes? A comparison study in a tertiary center in Turkey | Vural 2021 | x | | Turkey | 8103 Syrian refugees and 47151Turkish population | C-section rates were lower among refugee group |
| Vietnamese refugees in Adelaide: an obstetric analysis. | Ward 1981 | x | | Australia | 76 refugees | Vietnamese refugees had excellent antenatal care attendance, with no women missing more than two appointments, and even better attendance than the comparison group. A majority of women (91%) attended the 6-week postnatal appointment. Vietnamese refugees had significantly lower rates of C-sections and forceps deliveries compared to the comparison group. The need for analgesia was significantly lower among Vietnamese refugees compared to other groups. Primigravida Vietnamese refugees experienced a significantly shorter duration of the first phase of labor compared to the comparison group. |
| Perinatal outcomes of uninsured immigrant, refugee and migrant mothers and newborns living in Toronto, Canada. | Wilson-Mitchell 2013 | x | | Canada | 453 insured and uninsured migrant women | Uninsured women had a higher likelihood of receiving inadequate antenatal care. Rates of C-section and oxytocin augmentation exceeded provincial averages, with uninsured women being more likely to undergo a C-section due to issues with fetal heartbeats. |
| Evaluation of systems reform in public hospitals, Victoria, Australia, to improve access to antenatal care for women of refugee background: An interrupted time series design. | Yelland 2020 | x | | Australia | 1,341 migrant (English-speaking), 11,797 Migrant (non-English-speaking, and 2,740 Migrant with refugee background = | Both refugee women and Australian women showed similar trends in improvement regarding the recommended number of antenatal care visits. However, there was a consistent decline in the proportion of women who had their first visit during the first trimester. |

In the following section, we provide a summary of the main topics within the perinatal experience theme.

**Access to perinatal health services.** Access to adequate and timely prenatal care is a critical factor for the prevention and mitigation of adverse pregnancy outcomes [24–26]. In 2016, the World Health Organization's ANC model called for a minimum of eight ANC visits, with the first being within the first 12 weeks of gestation [27]. This can specifically be difficult for recently arrived refugees, who may not be aware of healthcare resources, may have limited

**Table 2. List of qualitative and mixed-methods studies on perinatal experiences of refugee/asylum-seeking women in OECD countries.**

| Title | Authors, year | Study Type | Refugees | Asylum Seekers | Host Country | Summary of findings |
|---|---|---|---|---|---|---|
| Avoiding obstetrical interventions among US-based Somali migrant women: a qualitative stu1dy. | Agbemenu 2021 | Qualitative | x | | United States | Some women intentionally avoided prenatal care due to privacy concerns, while others changed providers or hospitals out of fear of a cesarean section. These women often sought midwife care to reduce the likelihood of interventions and expressed distrust in medical technology. Some women experienced unwanted attention, including photographs taken by healthcare providers, which led them to feel disrespected and on display. Miscalculations in determining the last menstrual period sometimes caused confusion regarding due dates, further fueling distrust in technology. Many women believed they were treated differently due to their race, ethnicity, religion, or refugee status. |
| Maternal depression in Syrian refugee women recently moved to Canada: A preliminary study | Ahmed 2017 | mixed-methods | x | | Canada | Barriers to mental health services included social stigma, spousal refusal, and concerns about privacy and confidentiality. All women had a support person in their lives, mostly their partner, highlighting the importance of social and emotional support during pregnancy. Over half of the participants screened positive for depression, while half screened positive for PTSD. |
| Newly Arrived Migrant Women's Experience of Maternity Health Information: A Face-to-Face Questionnaire Study in Norway. | Bains 2021 | mixed-methods | | | Norway | Most refugees reported poor understanding of the information provided to them, while those who migrated for education or work had a better understanding. The need for an interpreter was highest during pregnancy, with 60% offered one, while only 20% were offered one during deliver |
| "They get a C-section... they gonna die": Somali women's fears of obstetrical interventions in the United States | Brown 2010 | Qualitative | x | | United States | Women experienced better access to perinatal care compared to Somalia but felt the need to seek care only when they were 5 months pregnant. They appreciated the improved access to emergency hospital services but felt rushed during delivery, leading to more interventions. They expressed fear towards having C-sections and associated them with death. |
| Resilience in international migrant women following violence associated with pregnancy | Gagnon 2014 | Qualitative | x | x | Canada | Women experienced challenges during pregnancy yet had lower rates of postpartum depression. Factors such as internal psychological support, coping mechanisms, social support, and government policies contributed to their resilience. |
| Psychosocial health of asylum seeking women living in state-provided accommodation in Germany during pregnancy and early motherhood: A case study exploring the role of social determinants of health | Gewalt 2018 | Qualitative | | x | Germany | Participants experienced fear and uncertainty due to their lack of experience and understanding of social norms. They faced challenges in obtaining social support because of the short time spent in reception centers and the need for relocation to other centers or cities. Many women expressed anxiety related to uncertainty and living in stressful environments. |

*(Continued)*

**Table 2.** (Continued)

| Title | Authors, year | Study Type | Refugees | Asylum Seekers | Host Country | Summary of findings |
|---|---|---|---|---|---|---|
| Access to Health Care for Pregnant Arabic-Speaking Refugee Women and Mothers in Germany. | Henry 2020 | Qualitative | x | | Germany | Women recognized the importance of antenatal care and had visits every 3–4 weeks. However, they faced knowledge gaps concerning healthcare services during pregnancy, childbirth, and pain management. Language barriers and the unavailability of interpreters contributed to anxiety and pressure, particularly during delivery. Gender-conforming care was important but not always prioritized. Access to detailed information about pregnancy and delivery from healthcare providers was limited, and being attended by unknown physicians or midwives raised concerns. Some women were critical of early pregnancy vaginal exams, while first-time mothers experienced uncertainty during the onset of labor. Support was sought from female relatives for advice, but language barriers continued to create anxiety and pressure for women, especially during childbirth. |
| Somali refugee women speak out about their needs for care during pregnancy and delivery. | Herrel 2004 | Qualitative | x | | United States | Participants believed that C-sections and episiotomies were more common in the US than in Somalia. They felt that interpreters lacked competence in medical terminology and questioned the staff's competence during delivery and postnatal care. Women experienced discrimination from nursing staff due to their race, color, and language barriers. Some were assigned interpreters without consultation, and participants reported a lack of cultural understanding by the nursing staff. |
| "I have to do what I believe": Sudanese women's beliefs and resistance to hegemonic practices at home and during experiences of maternity care in Canada. | Higginbottom 2013 | Qualitative | x | | Canada | Women believed that birth was a natural process and were hesitant to go to the hospital, fearing doctors would perform C-sections. Some women even resisted doctors' advice for a C-section. Additionally, women were reluctant to receive analgesia due to concerns about potential side effects. |
| Healthcare access challenges facing six African refugee mothers in South Korea: a qualitative multiple-case study. | Kim 2017 | Qualitative | x | x | Korea | Participants faced communication difficulties and language barriers, which led to information loss between them and healthcare providers. They felt unwelcome in health centers due to language and cultural barriers in South Korea. Many faced financial and economic challenges, as they lacked health insurance coverage, and free or subsidized ANC care was limited. Their legal status also hindered insurance access. Despite communication barriers, they acknowledged the high quality of Korean ANC and delivery services compared to their home countries. A lack of social support was particularly evident among early-stage asylum seekers. |

(*Continued*)

**Table 2.** (Continued)

| Title | Authors, year | Study Type | Refugees | Asylum Seekers | Host Country | Summary of findings |
|---|---|---|---|---|---|---|
| Factors Associated with the Presence of Strong Social Supports in Bhutanese Refugee Women During Pregnancy. | Kingsbury 2019 | mixed-methods | x | | United States | Social support mainly came from female family members and spouses. Women who relocated to another city were five times more likely to report high levels of social support, likely because they moved to reunite with family and friends. Connections established before settling in the US played a role in providing support. Receiving social support from close personal connections is considered a determinant of good maternal and child health. |
| Motherhood in the shade of migration: A qualitative study of the experience of Syrian refugee mothers living in Turkey | Korukcu 2018 | Qualitative | x | | Turkey | Language barriers impacted communication with healthcare professionals. However, most participants were satisfied with the quality of care they received. They experienced fears related to pregnancy and childbirth, feeling alone and separated from their community. Social support was limited due to isolation and language barriers, with participants expressing the emotional impact of leaving their mothers in Syria and lacking support. |
| Childbearing beliefs among Cambodian refugee women. | Kulig 1990 | Qualitative | x | | Canada | Women believed it was necessary to attend antenatal care appointments, but they had limited understanding of the purpose and efficacy of the tests and examinations they underwent during these appointments. |
| Reproductive health care for asylum-seeking women—a challenge for health professionals. | Kurth 2010 | mixed-methods | | x | Switzerland | The mode of delivery among the study group was not significantly different from the comparison group. Language barriers were a major concern, and even though interpreter services were available, finding an appropriate interpreter was not always easy. Asylum seekers faced a variety of problems, such as housing conditions, financing, contraception, and special diets, and sought help from hospital social services, authorities, and organizations. The lack of social support and stress often led to exhaustion among mothers. Many patients experienced violence prior to and after arrival, with more than half of them requiring some form of psychological care. Additionally, concerns about being forced to return to their country of origin contributed to their stress. |
| Refugee claimant women and barriers to health and social services post-birth. | Merry 2011 | Qualitative | x | | Canada | Women faced challenges in calling 911 during emergencies due to language barriers. There was a lack of psychological assessments, despite being at high risk. The unavailability of interpreters made it difficult for women to communicate and understand information. Many women were unsure where to access services. Separation from family and friends left them feeling isolated, especially for those who were new to the area. Over half of the participants experienced postpartum depression. |

(*Continued*)

**Table 2.** (Continued)

| Title | Authors, year | Study Type | Refugees | Asylum Seekers | Host Country | Summary of findings |
|---|---|---|---|---|---|---|
| The Personal Social Networks of Resettled Bhutanese Refugees During Pregnancy in the United States: A Social Network Analysis. | MKingsbury 2018 | Qualitative | x | | United States | Participants found that giving birth in the US was a less painful experience compared to their home country. They also highlighted the importance of social networks, mainly consisting of spouses and family connections, in sharing pregnancy-related anxieties. |
| The Experiences of African Women Giving Birth in Brisbane, Australia | Murray 2010 | Qualitative | x | | Australia | Women expressed fear of c-sections and believed that receiving analgesia would slow labor. Participants mentioned language as a barrier to care, often unaware of interpreter services, and had to rely on relatives and friends for help. Most preferred female doctors but consented to male doctors due to the Australian system. Patients faced difficulty navigating the Australian healthcare system and expressed frustration over the lack of continuity of care. While they generally found the quality of care to be good, some doctors or midwives were perceived as incompetent or inexperienced. Midwives occasionally asked inappropriate questions in front of interpreters or relatives. Participants expressed feelings of fear, loneliness, and uncertainty during their birth experiences. Some had family and community support but still felt that taking care of children was their primary responsibility. |
| Displaced mothers: birth and resettlement, gratitude and complaint. | Niner 2013 | Qualitative | x | | Australia | Due to the lack of interpreting services, some women faced difficulties in voicing their anxiety and postpartum hallucinations. Communication difficulties and confusion arose from the lack of interpreter services. Younger participants were more likely to voice complaints and express dissatisfaction with the services received. Some felt discriminated against and looked down upon by providers, with interactions varying along a spectrum. Women noted the lack of familiar social support, and some preferred delivery experiences within their own community in the camp. Additionally, some women experienced anxiety from events that had not been communicated to them. |
| Factors affecting implementation of perinatal mental health screening in women of refugee background. | Nithianandan 2016 | Qualitative | x | | Australia | All participants received perinatal mental health screening, with barriers to access primarily revolving around the presence of female interpreters and continuity of care. There was a need for appropriately translated screening tools and follow-up on positive symptoms. Access to in-person interpreters for perinatal mental health screening was highly encouraged over phone interpreters. Participants sometimes failed to disclose mental health symptoms in front of interpreters who were members of the local community. Continuity of care was highly emphasized for perinatal mental health, as it is critical for building trust. Social support played an important role in minimizing adverse mental health outcomes. |

*(Continued)*

**Table 2.** (Continued)

| Title | Authors, year | Study Type | Refugees | Asylum Seekers | Host Country | Summary of findings |
|---|---|---|---|---|---|---|
| Cultural background and socioeconomic influence of immigrant and refugee women coping with postpartum depression | O'Mahony 2013 | Qualitative | x | | Canada | Access to mental health services was challenging for many women due to social stigma and male partner's control and dominance. Financial difficulties and low socioeconomic status contributed to depression, as unemployment or work interfered with seeking support. Lack of social support contributed to postpartum depression in women. Refugee and immigrant women were more likely to experience postpartum depression due to financial, social, and cultural factors, as well as unemployment. |
| Perceptions of pregnancy experiences when using a community-based antenatal service: A qualitative study of refugee and migrant women in Perth, Western Australia. | Owens 2016 | Qualitative | x | | Australia | Women faced difficulties in attending hospital appointments due to inflexible scheduling and navigating the Australian system. The location of the facility facilitated easier access to services. Many women with limited English did not have interpreters available and relied on their husbands for interpretation. Participants preferred female providers and continuity of midwife care to avoid repeating information and receiving inconsistent advice. They also desired longer appointment times to ask questions and understand better. Many participants felt separated from extended family members and lacked their support. |
| Cultural safety and belonging for refugee background women attending group pregnancy care: An Australian qualitative study. | Riggs 2017 | Qualitative | x | | Australia | Women faced a lack of professional interpreters during childbirth, often relying on their husbands for interpretation. They also experienced dissatisfaction with privacy, as people entered without consultation and students performed procedures without considering their preferences. Women appreciated a care model that included bicultural workers and maternal child health nurses throughout their antenatal to postnatal journey. |
| Women from refugee backgrounds and their experiences of attending a specialist antenatal clinic. Narratives from an Australian setting. | Stapleton 2013 | mixed-methods | x | | Australia | Many women were unfamiliar with regular and prompt antenatal care (ANC) due to its impact on family life and transportation barriers. They were frustrated with labor induction, augmentation, and C-sections, considering them as western practices. Language barriers and the need for interpreters added challenges, while financial issues made scheduling frequent appointments difficult for high-risk women. |
| "COVID affected us all:" the birth and postnatal health experiences of resettled Syrian refugee women during COVID-19 in Canada. | Stirling Cameron 2021 | Qualitative | x | | Canada | Postnatal home services were sometimes canceled or unavailable, while accessing in-person interpreters posed challenges. Phone interpreters were preferred over having no interpretation, but in-person interpreters were better at understanding needs and body language. Pandemic restrictions left women feeling isolated and unable to seek informal support, such as having a doula or family members accompany them during delivery. |

(*Continued*)

**Table 2.** (Continued)

| Title | Authors, year | Study Type | Refugees | Asylum Seekers | Host Country | Summary of findings |
|---|---|---|---|---|---|---|
| Childbirth in exile: asylum seeking women's experience of childbirth in Ireland | Tobin 2014 | Qualitative | | x | Ireland | Language barriers and limited interpreter availability posed challenges, along with a lack of provider understanding of cultural backgrounds. Women experienced feelings of loneliness, isolation, and alienation. |
| Perinatal experiences of Somali couples in the United States. | Wojnar 2015 | Qualitative | | | United States | Language barriers and a lack of understanding of Somali culture by healthcare professionals led to difficulties, such as no gender-specific prenatal classes. Limited cultural understanding resulted in unmet expectations, mistrust, and doubt. Participants wished for better-educated and more respectful healthcare professionals. Men's participation was limited due to a lack of culturally adapted resources, and unexplained procedures caused anxiety and distrust among the women. |
| Birth Experience in Syrian Refugee Women in Turkey: A Descriptive Phenomenological Qualitative Study | Yaman Sözbir 2021 | Qualitative | x | | Turkey | Language barriers led to services not being adequately explained, and patients felt their privacy was not respected due to men entering the room and births occurring in multi-patient rooms. The service was described as quantitatively sufficient but unsatisfactory, with a lack of autonomy and a feeling of neglect. Women experienced neglect, prejudice, and felt healthcare personnel didn't spend enough time with them. Fear and anxiety were common in the delivery room, with a sense of loss of control. Consent was given without proper explanation, and women couldn't bring a hospital attendant to their room. |
| How do Australian maternity and early childhood health services identify and respond to the settlement experience and social context of refugee background families? | Yelland 2014 | Qualitative | x | | Australia | The majority of participants emphasized the need for professional interpreters during ANC visits, but few had access to them. Participants also expressed concerns about the gender of healthcare providers. Social hardship was reported by both women and men during the perinatal period, but maternity and early childhood services did not inquire about their social health. |
| Compromised communication: a qualitative study exploring Afghan families and health professionals' experience of interpreting support in Australian maternity care. | Yelland 2016 | Qualitative | x | | Australia | The majority of women in the study did not have access to professional interpreting services during labor, with many relying on husbands or family members with limited language proficiency. Interpreting services were mainly offered during booking visits, but not always available in subsequent visits. There was a preference for female care providers and female interpreters, but using interpreters from the local community created communication barriers as women were not comfortable sharing personal information. |

access to healthcare professionals and local clinics, and may lack transportation and childcare to attend ANC appointments. In this review, the timing and adequacy of antenatal care (ANC) visits among refugee/asylum-seeking women in OECD countries was reported in 28 articles. Most articles (64%, n = 18) found that refugees/asylum seekers were more likely to have a

delayed start of their ANC (after 12 weeks of gestation) and/or receive a fewer number of visits when compared with native-born or non-refugee populations. For example, Syrian refugees in Turkey had lower numbers of ANC visits compared to Turkish residents [28–31]. In Colombia, Venezuelan refugees had half the average number of ANC visits compared to Colombian women [32]. In Canada, immigrant women, including refugees, were more likely to start their ANC after three months of gestation compared to Canadian-born women [33]. In the Netherlands, all asylum-seeking women started their ANC after 13 weeks of gestation [34]. Similarly, a higher percentage of African and Hmong refugee women initiated their ANC later than women born in the United States [35,36].

Several factors can affect ANC utilization among pregnant women including residence, maternal age, maternal educational level, wealth quintile, access to mass media, the educational level of the husband, as well as health system policies [37–39]. In the present review, timing and adequacy of ANC (in terms of the recommended number of visits) among refugee/asylum seekers varied by the country of origin, the nature of displacement, residence (urban versus rural), and educational level of women. Specifically, one study found that immigrant women from humanitarian source countries (HSC) generally had late booking and poor ANC attendance (lower than the recommended number of visits) compared to immigrant women from non-HSC [40]. In Ireland, asylum seekers, on average, had a gestational age of 30 weeks at the time of booking [41]. In Sweden, asylum seekers and undocumented immigrants had even higher rates of inadequate ANC compared to refugees [42]. In Canada, asylum seekers also received inadequate ANC compared to other refugee groups [43]. Meanwhile, one study found no difference in the number of ANC visits between Syrian refugees and Turkish citizens; however, the authors attributed the finding to the rural nature of the setting and the higher educational level of Syrian refugees compared to Turkish residents [44]. Another study in Australia showed the number of antenatal care (ANC) visits improved among refugees in response to reforms of the public hospital system. However, this improvement came at the expense of their timely booking of care during the first trimester [45].

Several qualitative studies also highlighted the importance of refugee/asylum-seeking women's perspective towards ANC in understanding variations in ANC attendance [46–51]. For example, regular and prompt ANC was uncommon among refugee women in Australia because the scheduling of the appointments were seen as inconvenient and disruptive to their family life [47]. In another study, refugee women in Australia expressed the lack of flexible scheduling of hospital appointments which interfered with their ability to have regular ANC [48]. In the United States, Somali refugees held a cultural belief that care should be sought only after five months of gestation when the pregnant woman is clearly showing [49]. Another study, also on Somali refugees, found that women avoided ANC because they felt that it was a violation of their privacy [50]. In contrast, Arabic refugee women in Germany perceived the importance of perinatal care and had regular ANC visits every 3–4 weeks [51].

Compared to ANC, access to maternal postnatal health care or emergency hospital visits was infrequently examined within the studies. For example, only five papers discussed postnatal care (within the first 6 months after delivery) among refugee/asylum-seeking women in OECD countries [42,52–55]. Of these, only one early study reported high access among Vietnamese refugees in Australia between 1977 and 1980 [52]. In contrast, more recent papers highlighted difficulties seeking postnatal care between 2006 and 2020 in Canada [53], Sweden [42], and Germany [54]. Furthermore, one study highlighted that postpartum visits were cancelled or unavailable for resettled refugees in Canada owing to the COVID-19 pandemic [55]. However, the study did not assess whether native-born Canadian women had a similar experience during the pandemic.

Regarding access to emergency care, only two studies specifically examined this aspect of healthcare during pregnancy. In one study, Somali refugees in the United States expressed better access to emergency hospital services compared to Somalia [49]. The other study found that pregnant refugee women in Canada were unable to call 911 during emergencies due to language barriers [56]. Notably, little is known about refugee/asylum-seeking women's access to, utilization of, or perceptions about postnatal maternal emergency care services.

**Barriers to care.** Language barrier was the most frequently reported barrier among refugee/asylum-seeking women in OECD countries. Specifically, 25 studies unanimously highlighted language barriers and limited interpreting services in Australia [47,48,57–63], Canada [55,56], Germany [51,54,64], Ireland [41,65], Korea [66], Netherlands[67], Norway [68,69], Switzerland [70], Turkey [71,72], and the United States [73,74] as an impediment to optimal maternal care. For example, most Afghan refugees in Australia failed to access professional interpreting services during labor, interpretation services were mostly offered at booking visits with limited availability during follow-up visits, and interpretation was mostly done by husbands or family members with insufficient proficiency [62]. In Germany, women sometimes had to seek services outside the refugee clinic because of long waiting times where they had no interpreting services, had to be accompanied by neighbors, husbands or relatives for interpretation, or had to pay for out-of-pocket interpreter services. Some women in Germany also experienced anxiety during delivery due to language barriers [51]. In Canada, resettled Syrian refugees highlighted the challenge of accessing in-person interpretation services amid the COVID-19 pandemic as only phone interpretation was available, and sometimes it took multiple rounds of back and forth to provide health information given that doctors and interpreters were not available at the same time [55].

Refugee/asylum-seeking women in OECD countries frequently reported a preference for female healthcare providers and interpreters. The gender of healthcare providers and interpreters were particularly salient for women in the Australian studies [48,59,60,62]. Syrian refugees in Germany also expressed preferences to be examined only by female doctors [51,64].

Limited privacy was another theme in the studies. In Australia, refugee women felt that their privacy was invaded as medical students were brought in to observe and perform procedures without their approval [61]. In Turkey, Syrian refugees delivered in multi-patient rooms, and their privacy was not respected as many men entered the room without permission [72]. In the United States, Somali refugees felt that their privacy was not respected as healthcare providers photographed them or brought students and colleagues to observe medical procedures without their consent or prior knowledge [50]. Additionally, refugee women in Australia had difficulty disclosing mental health symptoms in front of interpreters who were members of the local community, potentially sharing sensitive information with others in the community [58].

Socioeconomic barriers and difficulty navigating the health system were also reported among refugee/asylum-seeking women in OECD countries. Specifically, several studies highlighted that women had financial difficulty [47,66,75] that impeded their healthcare access. In Germany, one study highlighted that low income was highly correlated with late onset of prenatal care [54]. Another study in Australia reported than Afghan refugees experienced financial and housing problems during the perinatal period; however, they were rarely asked about their socio-economic concerns during visits [59]. A study in Switzerland also indicated that asylum seekers, in particular, had a multitude of socioeconomic problems, and frequently needed help from hospital social services, authorities, and civil society organizations for housing and finances [70]. In addition, several studies indicated that refugee/asylum-seeking women had difficulty in navigating the healthcare system owing to its complexity, their inability to access health information or lack of information on where to seek services in Australia [48,60], Canada [56], and Germany [51]. In South Korea, legal status interfered with the

ability to have health insurance [66], and in Colombia, most Venezuelan refugees had no health insurance and faced barriers to care [32].

**Perception of service provision.** Eight papers reported findings related to the interaction of refugee/asylum-seeking women with healthcare, and most of them (88%, n = 7) highlighted negative interactions with healthcare providers. For example, in Australia, refugee/asylum-seeking women felt that they were discriminated against and looked down upon, they were being asked inappropriate questions in front of relatives, and physicians were less likely to ask about their general wellbeing [57,60]. Many Somali refugee/asylum-seeking women in the United States felt that their privacy was violated, as they were viewed with curiosity (interpreted as a form of voyeurism). Additionally, they felt disrespected and experienced differential treatment due to their color, religion, ethnicity, or inability to speak English. Furthermore, they felt that healthcare providers did not understand their cultural values and beliefs [50,73,74]. Similarly, refugee/asylum-seeking women felt neglected, disrespected, discriminated against, and misunderstood by healthcare providers in Ireland [65], Norway [68] and Turkey [72]. Meanwhile, only one study from Australia highlighted a positive experience at an antenatal care clinic with an open-door policy for refugees/asylum seekers [47].

Eleven papers highlighted refugee/asylum-seeking women's perceptions of care quality. Specifically, in Australia, refugee/asylum-seeking women felt that some staff were inexperienced; however, they were not always able to question their plan of care [60], and younger patients were more likely to voice their complaints and dissatisfaction with care (22). In another study, women preferred a longer appointment time to be able to ask questions and understand their plan of care [48]. Similarly, in the United States, Somali refugee/asylum- seeking women questioned the competence of staff during delivery and postnatal care [73], and expressed their distrust in the technology-based estimation of their expected delivery dates [50]. In Germany, Syrian refugees believed they were offered adequate prenatal care [54], while in another study, Arabic speaking refugees were dissatisfied with the lower frequency of ultrasound exams in Germany compared to their countries of origin [51]. In Turkey, Syrian refugee/asylum- seeking women described the perinatal services as being of good quality [71]. However, another study of Syrian refugees in Turkey found that the women perceived the care they received as sufficient but not satisfactory as they had little autonomy in their care plan and felt neglected [72]. Meanwhile, in Korea, refugee/asylum-seeking women expressed their satisfaction with Korean ANC and delivery services and considered them of good quality compared to the services available in their countries of origin. Finally, in Norway, women were generally satisfied with services [66]; however, having a Norwegian partner, higher education and high Norwegian language proficiency were associated with higher odds of dissatisfaction [68].

Continuity of care in terms of receiving continuous medical support during antenatal period, delivery, and postnatal period form the same caregiver is another critical component of healthcare quality and patient-centered approach to care [76]. We found seven papers that discussed the continuity of care among refugee/asylum-seeking women. Most studies were conducted in Australia, where the importance of continuous care was frequently emphasized [47,48,58,60,61]. For example, one study highlighted that continuous care helped to build trust, minimized the need to revisit traumatic events, minimized the time needed for interpreting services, and allowed patients more time to ask questions during consultations [47]. Another study indicated that continuity of care by a midwife was critical for the perinatal experience, as refugee/asylum-seeking women sometimes received different information when they asked the same questions to different providers [48]. One study noted that Karen women appreciated a model of care where they had a bicultural worker and a nurse from ANC to postnatal care [61]. Meanwhile, in Germany, Arabic-speaking refugee women expressed concern about being attended to by a midwife or a physician unknown to them [51].

**Experience during delivery.**   Interventions during delivery such as Cesarian-section (C-section), labor induction, and augmentation can mitigate several potential complications that may arise during the delivery process. However, routine and unnecessary administration of such interventions may negatively affect the birth process and result in more complementary interventions [77–79]. In the present review, between 1981 and 2021, 23 studies discussed interventions during delivery among refugee/asylum-seeking women in OECD countries. Specifically, nine studies reported lower rates of caesarean delivery (C-section), and/or labor induction among refugees/asylum-seeking women compared to host population or other immigrant groups in Australia [40,52,80], Turkey [28,29,31,81], United States [35], and the Netherlands [34]. Five studies found no difference in interventions during delivery between refugees/asylum-seeking women and comparison groups in Switzerland [70], Canada [82], Australia [83], Ireland [41], Turkey [44], and Germany. Two studies found that immigrants including refugees and asylum seekers had higher rates of C-section compared to the native-born population in Iceland [84] and Sweden [42], while one study highlighted that asylum seekers were more likely to have prior C-sections compared to other migrant groups [67].

In addition, six qualitative studies highlighted refugee/asylum-seeking women's perspectives towards delivery interventions. All studies reported that refugees and asylum seekers generally do not favor delivery interventions such as C-sections or labor induction. For example, studies from the United States revealed that Somali refugee/asylum-seeking women thought that C-sections were more practiced in the United States compared to their country of origin, felt they were rushed during the delivery process, expressed fear towards C-section and associated it with a higher risk of maternal death, expressed distrust in technology, sought care from only midwifes to avoid interventions, changed providers or hospitals because they were afraid of C-sections, and even intentionally delayed hospital arrival to avoid interventions [49,50,73]. In Australia, refugee/asylum-seeking women expressed fear towards C-sections, were frustrated with induction and augmentation of labor considering it a Western practice, and thought C-sections to be a premature medical decision [47,60]. Similarly, in Canada, refugee/asylum-seeking women believed that birth was a natural process; thus, some women resisted the medical advice of having a C-section and eschewed ANC at hospitals to avoid interventions [85].

Meanwhile, rates of analgesia during labor varied among refugee/asylum-seeking women. Specifically, one study from Canada found no difference in epidural/analgesia rates between migrant women, including refugees and native-born women [33], while another study highlighted that Sudanese refugees in Canada were reluctant to receive analgesia fearing side effects [85]. In Australia, early studies found lower rates for analgesia among Vietnamese refugees compared to other groups [52]. In more recent studies, refugee women in Australia believed that receiving analgesia would slow the labor process and they expressed frustration with what they perceived as healthcare providers' analgesia overenthusiasm [47,60]. One study also found that North African/Middle Eastern immigrants from non-HSCs in Australia were more likely to receive analgesia during labor; however, no significant difference was observed when compared to immigrants from HSCs [83]. Meanwhile, in the Netherlands, asylum-seeking women were more likely to receive analgesia during labor compared to the native-born population [34]. In the United States, some Bhutanese refugees expressed their satisfaction with analgesia and elaborated on how labor was a painless experience compared to their home country [86].

**Social support.**   Sixteen studies reported findings related to social support among refugee/asylum-seeking women between 2000 and 2017. More than half of the studies (56%, n = 9) described a lack of social support among refugee/asylum-seeing women in Australia [47,48,57], Canada [55,75], Turkey [71], Germany [87], Switzerland [70], and Korea [66]. One study from the United States also indicated that Somali men were not active participants in

social support during their partner's pregnancy due to the lack of culturally adapted sources engaging them [74]. Meanwhile, six studies indicated that social support during pregnancy was mostly offered by spouses, female relatives, or other extended family members, and highlighted its importance for navigating pregnancy-related anxieties and minimizing MH adversities [51,58,60,86,88,89].

**Perinatal mental health.** Perinatal mental health was a recurring theme in the literature among refugees/asylum-seeking women in the years between 2016 and 2021. Specifically, 11 studies described a sense of isolation, loneliness, alienation, fear and /or uncertainties among refugee/asylum-seeking women in Australia [47,48,60], Canada [55,56], Germany [51,87,90], Turkey [71,72] and Ireland [65]. Furthermore, five studies indicated that refugee/asylum-seeking women suffered or were at a higher risk of perinatal depression in Canada [56,75,88], Australia [47], and the United States [91]. In addition, eight studies highlighted that refugee/asylum-seeking women suffered anxiety, psychological distress or other mental health-related adversities during pregnancy in Germany [51,87,90,92], Australia [57,93], Switzerland [70] and the United States [74]. Only two studies found that despite psychological adversities, immigrant women including refugees/asylum seekers had lower or similar rates of postpartum depression compared to the native-born population in Canada [94,95].

Perinatal MH services were also examined in five papers [56–58,75,88,93]. Specifically, one paper highlighted that refugee women in Canada did not receive psychosocial assessment or support in the postpartum period despite being high risk (based on ad hoc notes by nurses and maternal reporting of services received) [56]. Other studies in Canada found that social stigma, spousal refusal, lack of privacy, and confidentiality were the main barriers for seeking perinatal MH services [75,88]. In Australia, lack of female interpreters, continuity of care, limited postpartum follow-up services, and lack of culturally adapted screening tools were identified as barriers to perinatal MH services [58]. For example, one study found that a lack of interpreting services interfered with refugee women's ability to voice their anxiety and post-partum hallucination complaints [57]. In addition, another study from Australia highlighted that the uptake of postnatal MH referral among refugees was double that for ANC, given that postnatal referral was offered after birth while patients were still at the hospital [93]. This suggests that both the need for and uptake of perinatal MH services among refugee women may be high, provided that barriers to access are addressed.

## Discussion

This scoping review has important policy implications as it sheds light on several adversities experienced by refugee/asylum-seeking women in OECD countries during the perinatal period. Access to timely prenatal care remains a challenge with a failure to address the root causes of the problem in several OECD countries, including those with a long history of hosting refugees. The limited availability of interpretation services and the lack of a patient-centered approach to care have also interfered with the perceived quality of care among refugee/asylum-seeking women. In addition, perceived isolation and the limited social support experienced by this vulnerable population have negatively impacted their perinatal experiences in OECD countries.

Findings from this review suggest that policy makers and organizations serving refugees and asylum seekers need to provide more information and support on how to navigate the healthcare system. Specifically, refugee/asylum-seeking women need more information on how to access services, communicate with providers, or schedule perinatal visits. Health facilities also need to offer a more flexible scheduling approach to accommodate refugee/asylum-seeking women's social needs and ensure the availability of adequate interpretation services.

Patient-centered approach to care is defined as "providing care that is respectful of and responsive to individual patient preferences, needs, and values and ensuring that patient values guide all clinical decisions" [96]. In this review, various deficiencies were identified in the patient-centered approach to maternity care. Specifically, in several countries, refugee/asylum-seeking women reported frequent encounters of disrespect, prejudice, difficulty in communication, lack of privacy, and limited involvement in their plan of care. Patient centered aspects to care such as dignity, respect, communication, and emotional support often influence the patients' perceived quality of care and patient satisfaction. Patient centered maternity care also extends beyond poor treatment to a more comprehensive, responsive and dignified approach to care as reflected in the WHO maternal and newborn health quality of care framework [97]. Although it may not always be feasible to accommodate refugee/asylum-seeking women's preferences for female healthcare providers, efforts should be made to ensure that patients fully understand and consent to the presence of medical students, residents and other healthcare staff being present during examinations or procedures.

Screening and management of mental health conditions during the perinatal period has been highlighted by various frameworks and guidelines [98–100]. Generally, almost one in five women can experience a mental health condition during the perinatal period, and among these, 20% can experience suicidal thoughts or self-harm [101]. Poor maternal mental health has been associated with adverse infant and child development [102]. Refugee and asylum-seeking women are especially vulnerable to experiencing perinatal mental health adversities [103]. In 2022, the WHO launched a guide focused on the integration of perinatal MH in routine maternal and child health services [104]. Given the findings of this review, stakeholders in OECD countries need to ensure that MH screening and referrals are provided as part of routine perinatal healthcare for refugee/asylum-seeking women.

As demonstrated by the global health literature and the results of this review, providing continuous maternity care by culturally trained medical staff can greatly improve refugee/asylum-seeking women's perinatal experiences, help build trust, allow more time for voicing fears and anxieties, give more space for patients to understand medical procedures, as well as improve perinatal mental health outcomes. Although having the same healthcare provider can be logistically challenging in health systems suddenly strained by refugee influxes, research shows that continuity of care can led to fewer interventions [105], improve women's satisfaction [106,107] and minimize unsafe situations that may arise from information loss and the inconsistency of advice among multiple caregivers [76,108].Thus, to the extent possible, healthcare systems providing care to refugee/asylum-seeking women should strive to provide continuous care by the same provider team for optimal maternal health outcomes.

From a policy perspective, perhaps the above findings call for the importance of a multidisciplinary model of refugee maternity care. This multi-disciplinary model would be specifically tailored towards refugee/asylum-seeking women needs, includes continuous perinatal care by culturally trained healthcare professionals, and incorporates both MH and social services components. In addition, this model of care would be flexible enough to consider the presence of competent interpreters and synchronize their availability with routine perinatal visits. A similar model of integrated care has been proposed by Correa-Velez et al [109] in Australia as a best practice model for refugee maternity care. In addition, Nujue et al [110] highlighted the importance of a multidisciplinary model for providing maternal healthcare for African refugee women in high income countries.

The results of this scoping review need to be interpreted in view of its limitations. First, we only included articles published till the end of 2021, of which only one paper [55] discussed the impact of the COVID-19 pandemic among refugee/asylum-seeking women. Likely, we have missed more recent articles documenting critical aspects of perinatal experience among

refugee/asylum-seeking women in OECD countries during the pandemic. In addition, at the time of writing, more than seven million refugees have fled Ukraine to neighboring OECD countries [111], which would change the ranking of OECD countries hosting refugees/asylum seekers. However, we still expect that many of the recommendations of this review are applicable to the Ukrainian refugee crisis, especially in terms of accessing timely integrated models of care for a better perinatal experience. Furthermore, we only focused on identifying common patterns experienced by refugee/asylum-seeking women in OECD countries without accounting for potential healthcare systems differences between the countries. Such structural variations may affect refugee/asylum-seeking women's access to and utlilization of healthcare services.

There is a need for further research to identify specific interventions that can improve the perinatal experiences of refugees/asylum seekers in OECD countries. Specifically, a mixed methods evaluation of refugee/asylum-seeking women experience utilizing different health system models including those with universal coverage of maternity services, systems with community-based perinatal care for refugees, as well as organizations offering specialized perinatal care clinics for refugee/asylum-seeking women. In addition, it will be important to consider the performance and resilience of these various models amid crises as the COVID-19 pandemic and the war in Ukraine.

## Conclusion

Refugee/asylum-seeking women experienced numerous adversities while seeking perinatal health services in OECD countries. Timely access to patient-centered maternity care and information on how to navigate the host country's healthcare system remain a top priority. Policy makers in OECD countries may need to consider the adoption of a multidisciplinary model of care that integrates both MH and social services with routine perinatal care, in the presence of competent and synchronized interpretation services.

## Supporting information

**S1 Checklist. Preferred Reporting Items for Systematic reviews and Meta-Analyses extension for Scoping Reviews (PRISMA-ScR) checklist.**
(DOCX)

**S1 File.**
(XLSX)

## Author Contributions

**Conceptualization:** Marwa Ramadan.

**Data curation:** Marwa Ramadan, Hani Rukh-E-Qamar.

**Formal analysis:** Marwa Ramadan, Hani Rukh-E-Qamar.

**Funding acquisition:** Zoua M. Vang.

**Investigation:** Marwa Ramadan.

**Methodology:** Marwa Ramadan.

**Project administration:** Marwa Ramadan, Zoua M. Vang.

**Resources:** Marwa Ramadan, Zoua M. Vang.

**Supervision:** Marwa Ramadan, Zoua M. Vang.

**Validation:** Marwa Ramadan.

**Visualization:** Marwa Ramadan.

**Writing – original draft:** Marwa Ramadan, Hani Rukh-E-Qamar, Zoua M. Vang.

**Writing – review & editing:** Marwa Ramadan, Seungmi Yang.

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
