## [Decision Letter · Decision Letter 0]

6 Mar 2023

PONE-D-23-02426Fifty years of evidence on perinatal experience among refugee and asylum-seeking women in Organization for Economic Co-operation and Development (OECD) countries: A scoping review

PLOS ONE

Dear Dr. Ramadan,

Thank you for submitting your manuscript to PLOS ONE. After careful consideration, we feel that it has merit but does not fully meet PLOS ONE’s publication criteria as it currently stands. Therefore, we invite you to submit a revised version of the manuscript that addresses the points raised during the review process.

Specifically these areas are in need of revision and improvement: 

Presentation of the data:

1. A separate table for qualitative studies

2. Provide informative tables

We look forward to receiving your revised manuscript.

Kind regards,

Forough Mortazavi

Academic Editor

PLOS ONE

Journal Requirements:

"This research was supported by Canadian Institutes of Health Research (CIHR) grant no. 159451 and a William Dawson Scholar Award to ZMV. The funders had no role in study design, data collection and analysis, decision to publish, or preparation of the manuscript."

"This research was supported by CIHR grant no. 159451 and a William Dawson Scholar Award to ZMV. The funders had no role in study design, data collection and analysis, decision to publish, or preparation of the manuscript."

Additional Editor Comments:

Thank you for submitting your paper to Plos One. PLS state why the authors preferred conducting a scoping review and not a systematic review or meta-analysis. What are the advantages of a scoping review vs. a systematic review or meta-analysis?

PLS describe how the data were analyzed. Enhance the supplementary file and include it as a table in the manuscript.

Reviewers' comments:

Reviewer's Responses to Questions

**Comments to the Author**

1. Is the manuscript technically sound, and do the data support the conclusions?

Reviewer #1: Partly

Reviewer #2: Yes

2. Has the statistical analysis been performed appropriately and rigorously? 

Reviewer #1: I Don't Know

Reviewer #2: No

3. Have the authors made all data underlying the findings in their manuscript fully available?

Reviewer #1: No

Reviewer #2: Yes

4. Is the manuscript presented in an intelligible fashion and written in standard English?

Reviewer #1: Yes

Reviewer #2: No

5. Review Comments to the Author

Reviewer #1: The authors need to motivate why it is important to focus on OECD countries, if using the authors stats on page 4 of the document 7.2 million out of 89.3 million displaced people are in OECD countries or 8% of all forcibly displaced migrants are in OECD countries

Why did the authors undertake a scoping review and not a systematic review?

How was the date range for included studies determined?

It is not clear what Figure 3 adds to the study (e.g. does it influence the interpretation of the results?)

For the reader to be able to better understand the results it is important to have a table with the author, dataset including sample size. time period and results as otherwise it is hard to interpret the findings as presented by the authors. There are also some inconsistencies in the text which could be made clearer with a table. For example on page 10, line 220 the authors mention Sweden and Netherland on line 199 but then on page 8 line 155 the Netherlands and Sweden are not mentioned as countries where studies were conducted

A separate table could be made for qualitative studies.

Proofread sentence on Page 11 line 236

In the Barrier to care section beginning on page 12. there is some repetition around privacy-these papers should all be brought together and discussed once

Page 14: Perceptions of care, it may be helpful to link some of the findings to privacy as that also comes up here around voyeurism

In terms of the qualitative data, how much of the findings are unique to asylum seekers and refugees and how much is a reflection of the health care system?

Reviewer #2: Thank you for submitting your paper in Plus one.

The results of this paper is not to important to publish in this journal.

you can revise your study and with a suitable analysis such as meta analysis resubmit that.

6. PLOS authors have the option to publish the peer review history of their article (what does this mean?). If published, this will include your full peer review and any attached files.

Reviewer #1: **Yes: **Heather Brown

Reviewer #2: No

---

## [Author Response · Author response to Decision Letter 0]

28 Apr 2023

Response to comments

On behalf of the authors, I would like to thank the editor and all the reviewers for their comments and suggestions to improve the quality of the manuscript. We took all comments into consideration and addressed them to the extent possible. Below is a point by point response to the comments and the suggestions by the academic editor and the reviewers including a description of the changes done to the older version of the manuscript. [also attached as a separate file as part of submission documents]

Editor comments 

Our response: Done, we reviewed the manuscript to meet PLOS ONE style requirements 

Please remove any funding-related text from the manuscript and let us know how you would like to update your Funding Statement. Currently, your Funding Statement reads as follows: "This research was supported by CIHR grant no. 159451 and a William Dawson Scholar Award to ZMV. The funders had no role in study design, data collection and analysis, decision to publish, or preparation of the manuscript." Please include your amended statements within your cover letter; we will change the online submission form on your behalf 

Our response: Done-The funding statement was removed from the manuscript body. 

No further changes are required to the funding statement in the online submission form.

3.PLS state why the authors preferred conducting a scoping review and not a systematic review or meta-analysis. What are the advantages of a scoping review vs. a systematic review or meta-analysis?

Our response: Done- Page 6 in the Updated Manuscript. The following statement were added to the methods section “In this study, we chose to conduct a scoping review instead of a systematic review or meta-analysis because our primary objective was to map the existing evidence on perinatal health experiences among refugees and asylum seekers in OECD countries. Given the wide search range, the heterogeneous nature of the available evidence, and the complex aspects of perinatal experiences that involve moving beyond the quantitative examination of evidence, a scoping review was better suited to meet our research goals.”

4.PLS describe how the data were analyzed. 

Our response: Done- a descriptive and a narrative approach was followed for data analysis. Methods section was expanded in the manuscript to provide a more detailed stepwise approach on how data analysis was done as indicated by track changes in the updated manuscript [Pages 6, 7, 8]

5.Enhance the supplementary file and include it as a table in the manuscript.

Our response: Done- Two tables were added to the main body of the manuscript, one for quantitative studies and one for qualitative and mixed-methods studies. Extra columns [including sample sizes (for) quantitative studies and summarized findings] were added to enhance the understanding and interpretation of results as recommended by the editors and reviewers 

 

Reviewer 1

6. The authors need to motivate why it is important to focus on OECD countries, if using the authors stats on page 4 of the document 7.2 million out of 89.3 million displaced people are in OECD countries or 8% of all forcibly displaced migrants are in OECD countries 

Our response: Done- we modified the following sentences in the objective paragraph [Page 5 in the updated manuscript ] as follows: 

“Previous reviews have concentrated on documenting adverse pregnancy outcomes among immigrant women, including refugees and asylum seekers (8,14–16). However, to our knowledge, there has been no documentation of the experiences of refugee/asylum seeking women with perinatal health services in OECD countries. As there has been a significant increase in humanitarian migration to OECD countries over the past decade, and with their commitment towards immigrants and refugees as part of the Sustainable Development Goal 3 (17,18), it is imperative from a policy perspective to comprehend the various challenges faced by this vulnerable group for better integration, planning, and allocation of resources. Hence, this scoping review aims to describe the general trends in perinatal health research among refugees/asylum seekers in OECD countries from 1970 to 2021 and primarily focuses on summarizing the findings related to their perinatal experience.”

7. Why did the authors undertake a scoping review and not a systematic review? 

Our response: Our primary objective was mapping the available evidence on perinatal experience among refugees/asylum seekers in OECD countries in the past 50 year so a scoping review was a more realistic approach compared to a systematic review or meta-analysis [which tend to be more specific and focused]. In addition, the authors wanted to be as comprehensive as possible in describing perinatal research trends, so we included different study designs [quantitative qualitative, and mixed methods] which would have been challenging in systematic reviews or meta-analyses. 

For clarity: we included the rationale for conducting a scoping review instead of a systematic review or met analysis in the main body of the manuscript [Page 6] in the updated manuscript] as follows:

“In this study, we chose to conduct a scoping review instead of a systematic review or meta-analysis because our primary objective was to map the existing evidence on perinatal health experiences among refugees and asylum seekers in OECD countries. Given the wide search range, the heterogeneous nature of the available evidence, and the complex aspects of perinatal experiences that involve moving beyond the quantitative examination of evidence, a scoping review was better suited to meet our research goals”

8.Justficiation for date range:

Our response: We provided the following statement [Page 5 in the updated manuscript] to explain the rationale behind the selection of the date range:

“This interval was selected to provide a historical and comprehensive perspective on the evolution of perinatal experiences among refugees and asylum seekers in OECD countries. It facilitated the examination of trends in perinatal health research in the context of major global displacement events, as well as the progress in perinatal care and research methodologies over the past five decades. This extensive range also allowed us to explore a wide variety of research, uncovering patterns and presenting a thorough overview of perinatal experiences for refugees and asylum seekers in OECD nations.”

In addition, electronic databases were more readily available 1970 onwards and since the search was conducted in March 2022, we decided to set the range between 1970 and 2021

9.It is not clear what Figure 3 adds to the study (e.g. does it influence the interpretation of the results?)

Our response: As noted in our background section, in this scoping review, “we aimed to describe the general trends in perinatal health research among refugees/asylum seekers in OECD countries over the fifty years from 1970 to 2021 and primarily focuses on summarizing the findings related to their perinatal experience.”

Therefore, we wanted to explore how the general trends in perinatal health research (including perinatal experience) have been historically linked to major displacement crises. The figure would not necessarily influence the interpretation of findings related to perinatal experience, but it can help the readers historically contextualize the trends in perinatal research among refugees/asylum seekers as one of our objectives 

10. For the reader to be able to better understand the results it is important to have a table with the author, dataset including sample size. time period and results as otherwise it is hard to interpret the findings as presented by the authors. There are also some inconsistencies in the text which could be made clearer with a table. For example on page 10, line 220 the authors mention Sweden and Netherland on line 199 but then on page 8 line 155 the Netherlands and Sweden are not mentioned as countries where studies were conducted

A separate table could be made for qualitative studies. 

Our response: Done- two Tables were added to the main body of the manuscript, one for quantitative studies and one for qualitative. Extra columns [including sample sizes and summarized findings] were added to enhance the understanding and interpretation of results as recommended by the editors and reviewers.

Regarding Sweden and Netherlands, it is correct that it was not mentioned in line 155 (older manuscript) as we only mentioned the top five host countries which were Australia (18%, n = 15), Turkey (17%, 155 n = 15), Canada (16%, n = 13), Unites States (12%, n = 10), and Germany (10%, n = 8). Only 2 studies were conducted in Sweden and 3 studies were conducted in the Netherlands, therefore they were not listed among the top five countries. Later in the text, when we described the perinatal experience findings, we listed the host country for each study. Therefore, both Netherlands and Sweden were listed. 

For clarity and following the editor recommendation, we moved the supplementary table to the main body of manuscript and host country is currently listed as a column for each study on perinatal experience. 

11. Proofread sentence on Page 11 line 236 

Our response: Done, the sentence was revised to “Regarding access to emergency care, only two studies specifically examined this aspect of healthcare during pregnancy”

12. In the Barrier to care section beginning on page 12. there is some repetition around privacy-these papers should all be brought together and discussed once

Page 14: Perceptions of care, it may be helpful to link some of the findings to privacy as that also comes up here around voyeurism

Our response: Done, text was edited in both sections for more clarity as follows:

Page 12 [old version] privacy paragraph was changed to the following: 

Privacy was also an issue, as refugee women in Australia felt that their privacy was invaded as medical students were brought in to observe and perform procedures without their approval (56). In Turkey, Syrian refugees delivered in multi-patient rooms, and their privacy was not respected as many men entered the room without permission (68). In the United States, Somali refugees felt that their privacy was not respected as healthcare providers photographed them or brought students and colleagues to observe medical procedures without their consent or prior knowledge (45). Additionally, refugee women in Australia had difficulty disclosing mental health symptoms in front of interpreters who were members of the local community, potentially sharing sensitive information with others in the community (53).

Page 14 [old version]: paragraph was edited to the following: 

Many Somali refugee/asylum seeking women in the United States felt that their privacy was violated as they were viewed with curiosity (as a form of voyeurism). Additionally, they felt disrespected and experienced differential treatment due to their color, religion, ethnicity, or inability to speak English. 

12. In terms of the qualitative data, how much of the findings are unique to asylum seekers and refugees and how much is a reflection of the health care system? 

Our response: We would like to thank the reviewer for this very interesting question. Throughout the themes and subthemes explored in this manuscript, we believe the findings represent a mix of factors that are unique to asylum seekers and refugees as well as those reflecting the healthcare system. However, it was difficult to determine an exact proportion of each aspect within the findings, as the extracted findings were more reflective of the interaction of these factors in shaping the perinatal experience. 

Reviewer #2: 

13. Thank you for submitting your paper in Plus one.

you can revise your study and with a suitable analysis such as meta-analysis resubmit that. We would like to thank reviewer #2 for his feedback to improve the quality of the submitted manuscript. 

Our response: We agree that metanalysis is a stronger design compared to a scoping review; however, it would not have been appropriate for the present study. Specifically, our primary objective was mapping the available evidence on perinatal experience among refugees/asylum seekers in OECD countries in the past 50 year, so we needed a more inclusive design. Therefore, scoping review was a more realistic and efficient approach compared to a systematic review or meta-analysis [that tend to be more specific and focused]. In addition, the authors wanted to be as comprehensive as possible in describing research trends, so we included different types of study designs [quantitative, qualitative, and mixed methods] which would be challenging in systematic reviews or meta-analyses that necessitate a stringent selection criterion for studies 

For clarity, we included the rationale for conducting a scoping review instead of a systematic review or metanalysis [Page 6 in the updated manuscript] as follows: “In this study, we chose to conduct a scoping review instead of a systematic review or meta-analysis because our primary objective was to map the existing evidence on perinatal health experiences among refugees and asylum seekers in OECD countries. Given the wide search range, the heterogeneous nature of the available evidence, and the complex aspects of perinatal experiences that involve moving beyond the quantitative examination of evidence, a scoping review was better suited to meet our research goals.”

In addition, to enhance the understanding of results, we expanded our tables to include the summarized findings for the included quantitative and qualitative studies. Specifically, two tables were added to the main body of the manuscript, one for quantitative studies and one for qualitative. Extra columns [including sample sizes and summarized findings] were added to enhance the understanding and interpretation of results as recommended by the editors and reviewers

---

## [Decision Letter · Decision Letter 1]

11 May 2023

PONE-D-23-02426R1Fifty years of evidence on perinatal experience among refugee and asylum-seeking women in Organization for Economic Co-operation and Development (OECD) countries: A scoping reviewPLOS ONE

Dear Dr. Ramadan,

Thank you for submitting your manuscript to PLOS ONE. After careful consideration, we feel that it has merit but does not fully meet PLOS ONE’s publication criteria as it currently stands. Therefore, we invite you to submit a revised version of the manuscript that addresses the points raised during the review process.

We look forward to receiving your revised manuscript.

Kind regards,

Forough Mortazavi

Academic Editor

PLOS ONE

Journal Requirements:

Additional Editor Comments:

Thank you for revising the manuscript. It has been improved but there are still some minor issues raised by the reviewer which need to be addressed.

Reviewers' comments:

Reviewer's Responses to Questions

**Comments to the Author**

1. If the authors have adequately addressed your comments raised in a previous round of review and you feel that this manuscript is now acceptable for publication, you may indicate that here to bypass the “Comments to the Author” section, enter your conflict of interest statement in the “Confidential to Editor” section, and submit your "Accept" recommendation.

Reviewer #1: (No Response)

2. Is the manuscript technically sound, and do the data support the conclusions?

Reviewer #1: Yes

3. Has the statistical analysis been performed appropriately and rigorously? 

Reviewer #1: N/A

4. Have the authors made all data underlying the findings in their manuscript fully available?

Reviewer #1: Yes

5. Is the manuscript presented in an intelligible fashion and written in standard English?

Reviewer #1: Yes

6. Review Comments to the Author

Reviewer #1: Thank you for your response to feedback. I think the paper is much improved however, there are still some minor issues which need to be addressed.

On page 5, line 84, you mention that no research has been done on asylum seekers and refugees in OECD countries. This strictly is not correct: See Heslehurst, N., Brown, H., Pemu, A., Coleman, H., & Rankin, J. (2018). Perinatal health outcomes and care among asylum seekers and refugees: a systematic review of systematic reviews. BMC medicine, 16(1), 1-25. which is an umbrella review across high income countries which includes OECD countries. There should be additional motivation or state what this study adds in addition to this existing body of research

I also think that figure 3 should be removed as it is not clear how it contributes to the discussion.

7. PLOS authors have the option to publish the peer review history of their article (what does this mean?). If published, this will include your full peer review and any attached files.

Reviewer #1: No

---

## [Author Response · Author response to Decision Letter 1]

25 May 2023

Response to comments

On behalf of the authors, I would like to thank the editor and all the reviewers for their comments and suggestions to improve the quality of the manuscript. We took all comments into consideration and addressed them within the manuscript. Below is a point-by-point response to the latest comments and the suggestions by the reviewer 1 including a description of the changes made to the older version of the manuscript.

Please review your reference list to ensure that it is complete and correct. If you have cited papers that have been retracted, please include the rationale for doing so in the manuscript text or remove these references and replace them with relevant current references. Any changes to the reference list should be mentioned in the rebuttal letter that accompanies your revised manuscript. If you need to cite a retracted article, indicate the article’s retracted status in the References list and also include a citation and full reference for the retraction notice. 

Our response: Done 

Reviewer #1

comment 1 

Thank you for your response to feedback. I think the paper is much improved however, there are still some minor issues which need to be addressed.

On page 5, line 84, you mention that no research has been done on asylum seekers and refugees in OECD countries. This strictly is not correct: See Heslehurst, N., Brown, H., Pemu, A., Coleman, H., & Rankin, J. (2018). Perinatal health outcomes and care among asylum seekers and refugees: a systematic review of systematic reviews. BMC medicine, 16(1), 1-25. which is an umbrella review across high income countries which includes OECD countries. There should be additional motivation or state what this study adds in addition to this existing body of research

Our response: We would like to thank the reviewer for his insightful comment and for drawing our attention to the paper by Heslehurst et al. (2018). We acknowledge that the cited paper included a general overview of some aspects of perinatal experience across refugees/asylum seekers in high income countries, which included some OECD countries. However, it did not deeply dive into the specific details of these experiences. 

To be more accurate and address the points raised by the reviewer: we revised the two paragraphs at the end of background section to the following: 

Previous reviews have primarily focused on documenting adverse pregnancy outcomes among immigrant women, including refugees and asylum seekers (8,14–16). However, only a few studies have examined aspects related to their perinatal experience. For example, a comprehensive study by Heslehurst et al. (8) performed an umbrella review of perinatal health outcomes and care among refugees and asylum seekers, with some OECD countries represented in their sample. While the latter provided an insightful general overview of perinatal access and experiences among migrant women, the specific aspects of these experiences among refugees and asylum-seeking women in OECD countries are yet to be deeply explored.

 Given the significant increase in humanitarian migration to OECD countries over the past decade, coupled with the organization’s commitment towards immigrants and refugees as part of the sustainable development goal 3 (17,18), it is imperative from a policy perspective to understand the various challenges faced by this vulnerable group for better integration, planning, and allocation of resources. Therefore, this scoping review aims to expand upon the existing literature by describing the general trends in perinatal health research among refugees/asylum seekers in OECD countries from 1970 to 2021 and synthesize the specific findings related to their perinatal experiences over the specified period.

comment 2 

I also think that figure 3 should be removed as it is not clear how it contributes to the discussion. 

Our response : The referenced figure was removed as recommended by the reviewer [Indicated by track changes]

---

## [Decision Letter · Decision Letter 2]

8 Jun 2023

Fifty years of evidence on perinatal experience among refugee and asylum-seeking women in Organization for Economic Co-operation and Development (OECD) countries: A scoping review

PONE-D-23-02426R2

Dear Dr. Ramadan,

We’re pleased to inform you that your manuscript has been judged scientifically suitable for publication and will be formally accepted for publication once it meets all outstanding technical requirements.

Kind regards,

Forough Mortazavi

Academic Editor

PLOS ONE

Additional Editor Comments (optional):

Reviewers' comments:

Reviewer's Responses to Questions

**Comments to the Author**

1. If the authors have adequately addressed your comments raised in a previous round of review and you feel that this manuscript is now acceptable for publication, you may indicate that here to bypass the “Comments to the Author” section, enter your conflict of interest statement in the “Confidential to Editor” section, and submit your "Accept" recommendation.

Reviewer #1: All comments have been addressed

2. Is the manuscript technically sound, and do the data support the conclusions?

Reviewer #1: Yes

3. Has the statistical analysis been performed appropriately and rigorously? 

Reviewer #1: N/A

4. Have the authors made all data underlying the findings in their manuscript fully available?

Reviewer #1: Yes

5. Is the manuscript presented in an intelligible fashion and written in standard English?

Reviewer #1: Yes

6. Review Comments to the Author

Reviewer #1: Authors have addressed all my comments.

There is nothing else left to address. Authors may consider a final re-read to fix any minor grammatical errors

7. PLOS authors have the option to publish the peer review history of their article (what does this mean?). If published, this will include your full peer review and any attached files.

Reviewer #1: No

---

## [Editor Report · Acceptance letter]

15 Jun 2023

PONE-D-23-02426R2 

Fifty years of evidence on perinatal experience among refugee and asylum-seeking women in Organization for Economic Co-operation and Development (OECD) countries: A scoping review 

Dear Dr. Ramadan:

I'm pleased to inform you that your manuscript has been deemed suitable for publication in PLOS ONE. Congratulations! Your manuscript is now with our production department. 

Kind regards, 

on behalf of

Dr. Forough Mortazavi 

Academic Editor

PLOS ONE